# Non-random host tree infestation by the Neotropical liana *Marcgravia longifolia*

Eckhard W. Heymann[1], Sarina Thiel[2], Filipa Paciência[1,3], Milagros N. Rimachi Taricuarima[4], Ricardo Zárate Gómez[5], Ney Shahuano Tello[6], Katrin Heer[2,7], Holger Sennhenn-Reulen[8,9,10] and Roger Mundry[8,9,11]

[1] Verhaltensökologie & Soziobiologie, Deutsches Primatenzentrum –Leibniz-Institut für Primatenforschung, Göttingen, Göttingen, Germany
[2] Conservation Ecology, Department of Biology, Philipps Universität, Marburg, Germany
[3] Departamento de Biologia Animal, Faculdade de Ciências, Universidade de Lisboa, Lisbon, Portugal
[4] Facultad de Ciencias Biológicas, Universidad Nacional de la Amazonía Peruana, Iquitos, Peru
[5] Instituto de Investigaciones de la Amazonía Peruana (IIAP), Iquitos, Peru
[6] Estación Biológica Quebrada Blanco, Río Tahuayo, Loreto, Peru
[7] Forest Genetics, Faculty of Environment and Natural Resources, Albert-Ludwigs-Universität Freiburg, Freiburg, Germany
[8] Cognitive Ethology Laboratory, Deutsches Primatenzentrum—Leibniz-Institut für Primatenforschung, Göttingen, Germany
[9] Leibniz ScienceCampus Primate Cognition, Göttingen, Germany
[10] Nordwestdeutsche Forstliche Versuchsanstalt, Göttingen, Germany
[11] Department of Primate Cognition, Georg-August-University Göttingen, Göttingen, Germany

Corresponding author
Eckhard W. Heymann,
eheyman@gwdg.de

## ABSTRACT

The question whether or not tropical lianas infest host trees randomly or they exert host selection has implications for the structure and dynamics of tropical rainforests, particularly if colonization by lianas impacts host fitness. In this study, we present evidence that the Neotropical liana *Marcgravia longifolia* (Marcgraviaceae) infests host trees non-randomly. We identified host trees to species or genus level for 87 of the 100 *M. longifolia* individuals found in the study area of the Estación Biológica Quebrada Blanco (EBQB) in north-eastern Peruvian Amazonia. Data on host availability were taken from two 1-ha plots sampled at EBQB as part of a large-scale tree inventory in western Amazonia. Of the total of 88 tree genera with two or more individuals present in the inventory, 18 were represented amongst hosts. Host genera with a probability of colonization higher than expected by chance were *Eschweilera* (Lecythidaceae), *Pouteria* (Sapotaceae), *Brosimum* (Moraceae), and *Hymenaea* (Fabaceae). These findings suggest that *M. longifolia* exerts some level of host selectivity, but the mechanisms for this are completely unknown. Given the large number of animal species (41 bird species, three primate species) that are dispersing the seeds of *M. longifolia* and that have diverse ecological strategies, directed seed dispersal is unlikely to account for the observed patterns of host infestation.

## INTRODUCTION

Negative impacts of liana infestation on tree growth, reproduction and survival, and alteration of forest dynamics may significantly decrease the carbon storage potential of tropical ecosystems (*García León et al., 2018*; *Ingwell et al., 2010*; *Kainer et al., 2006*; *Laurance et al., 2014*; *Schnitzer & Bongers, 2011*; *van der Heijden et al., 2013*). Despite their ecological importance, the ecology of lianas has been much less studied than that of trees, although considerable progress has been made in recent years (*Schnitzer, Mangan & Hubbell, 2015*). An important aspect for forecasting future impacts on forest dynamics is the question of whether or not lianas prefer or avoid specific host tree species.

Infestation patterns have been mainly addressed from the perspective of trees, *i.e.,* which proportion of the tree community is infested and how severe infestations are. Fast-growing tree species that are tall as adults are less likely to support lianas (*van der Heijden, Healey & Phillips, 2008*; *Sfair et al., 2016*), and light-demanding tree species have lower liana prevalence (*Visser et al., 2018*). Trees with larger diameter at breast height (dbh) support more lianas and a higher liana biomass (*Pérez-Salicrup & de Meijere, 2005*). In some studies, host tree traits such as bark roughness and flakiness, emerged as factors influencing infestation (*van der Heijden, Healey & Phillips, 2008*; *Carsten et al., 2002*), while in other studies factors related to light and dispersal seemed to be more important than host tree traits (*Malizia & Grau, 2006*). Finally, there are habitat differences (tropical rainforest, savanna woodland, tropical semi-deciduous forest) in the interaction between lianas and hosts, possibly structured by morphological traits (*e.g.*, bark characteristics) of host trees (*Zulqarnain Silva et al., 2016*). However, the question whether lianas prefer or avoid certain tree species as hosts has rarely been addressed. In an Amazonian forest some tree species were less infested than would be expected from the mean percent of infestation (*Pérez-Salicrup, Sork & Putz, 2001*). In a study in the Indian Eastern Ghats, a few tree species were more strongly infested than expected from their abundance (*Chittibabu & Parthasarathy, 2001*), and in northern temperate forests some liana species seem to preferentially infest certain tree species (*Leicht-Young et al., 2010*). In regenerating deciduous forests in the Piedmont region of New Jersey (USA), four of the five most abundant lianas preferred early successional trees (*Ladwig & Meiners, 2010*). Finally, in regenerating lowland rainforest in Ile-Ife (Nigeria) three liana species preferentially infested specific tree species (*Uwalaka, Borisade & Rufai, 2021*).

In this article, we examine liana-host tree associations from the perspective of the liana, *i.e.,* we test whether a specific liana species infests hosts according to their availability or whether there are host preferences and avoidances, respectively. Our model system is the woody liana *Marcgravia longifolia* from the Neotropical family Marcgraviaceae. *Marcgravia longifolia* is unusual within the family by presenting long pedunculate and flagelliflorous cauligenous inflorescences and infructescences that arise from the unbranched stem from ground level up to the canopy (Fig. 1A). Nectaries attract bats and hummingbirds, and a large number of frugivorous animals (birds, primates, and bats) feed on the fruits and disperse the seeds (*Tirado Herrera et al., 2003*; *Domingues Paciência, 2014*; *Willems, 2016*). *Marcgravia longifolia* individuals are rooted close to the base of their host trees; juveniles

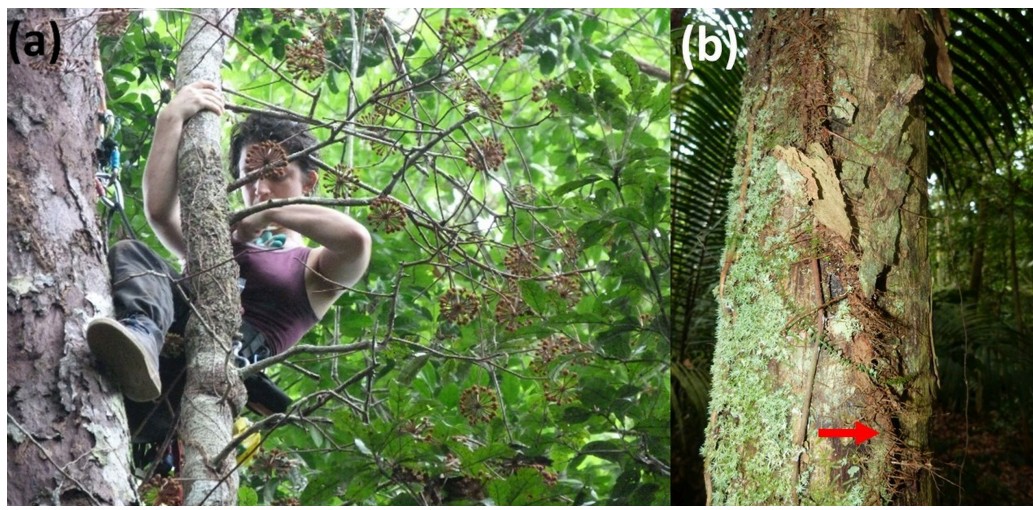

**Figure 1** **Growth habit of *Marcgravia longifolia*.** (A) The cauligenous infructescences of *Marcgravia longifolia* are present all along the trunk from ground level to canopy level. (B) A *Marcgravia longifolia* individual (red arrow) growing on the trunk of an *Eschweilera coriacea* tree with strongly exfoliating bark.

creep up trunks and attach by adventitious roots (*Heald, de Roon & Dressler, 2002*). During our research on the plant-animal interactions of *M. longifolia*, we got the impression that this liana is found more frequently on some host trees than on others. Therefore, we set out to systematically test the hypothesis that *M. longifolia* infests certain host tree species more frequently than would be expected from tree availability against the null hypothesis that infestation is random.

## METHODS

### Study site

The study was conducted from October 2013 to January 2014 at the Estación Biologica Quebrada Blanco (EBQB), located in north-eastern Peruvian Amazonia (4°21′S, 73°09′W), 70 km southeast of Iquitos. Primary terra-firme forest ("bosque de altura"; *Encarnación, 1985*) interspersed with small swampy areas ("bajiales"; *Encarnación, 1985*) dominates the 100 ha study area which is equipped with a trail system with 100 m ×100 m grids. Annual rainfall is around 3000 mm with a peak from February to May and a trough between August and October. For further details of EBQB see *Heymann & Tirado Herrera (2021)* and *Heymann, Tirado Herrera & Dolotovskaya (2021)*.

### Data collection

Since *M. longifolia* is a fruit resource for several primate species (*Tirado Herrera et al., 2003*), many *M. longifolia*-individuals (and their GPS position) were known from continuous primate ecological field work at EBQB. To detect additional individuals, we employed two strategies: (1) Systematic transect walks along all trails of the trail system throughout the study area. This allowed detection of reproductive *M. longifolia*-individuals within 25-30 m from the trails. (2) Systematic search with in the 1 ha-cells of the trail grid system,

to detect individuals not visible from the trails. In combination, this resulted in a full inventory of reproductive *M. longifolia*-individuals in the study area. For all individuals the GPS position was recorded. Detection of *M. longifolia* is facilitated by their unique growth habit with inflorescences and infructescences protruding from the trunk (Fig. 1A) which can be seen from a long distance, particularly during the fruiting period. We identified *M. longifolia* host trees using a regional flora (*Vásquez Martínez, 1997*). All except one *M. longifolia* individual (found in a "bajial") were growing in terra-firme forest ("bosque de altura") with a flat or only slightly undulating terrain ("bosque de terraza"; *Encarnación, 1985*). No *M. longifolia* individuals were found in those parts of the study area with strongly undulating terrain ("bosque de colina"; *Encarnación, 1985*). All field work was performed under permits from the Peruvian Ministry of Agriculture in Lima (permits 0073-2014-MINAGRI-DGFFS/DGEFFS, 304-2018-MINAGRI-SERFOR-DGGSPFFS, 445-2018-MINAGRI-SERFOR-DGGSPFFS, 528-2019-MINAGRI-SERFOR-DGGSPFFS).

## Data analyses

To estimate the extent to which *M. longifolia* selected host tree species randomly, *i.e.,* according to availability, or preferred/avoided certain species, respectively, and the extent to which such potential preferences differed between genera, we first compiled a list of tree species and their abundance at EBQB. Data were taken from an inventory of trees >10 cm dbh in two 1-ha plots (a total of 1,087 trees) by *Dávila Cardozo & Ríos Paredes (2006)* which was part of a large-scale study on Amazonian tree communities (*Pitman et al., 2008*), (*ter Steege et al., 2013*). These two plots are located in terra-firme forest ("bosque de altura") of the "bosque de terraza"-type (*Encarnación, 1985*), *i.e.,* more or less flat terrain, and without swampy areas; thus, they are representative for the type of habitat where we found *M. longifolia*. As the number of tree species (294 species; *Dávila Cardozo & Ríos Paredes, 2006*) was much larger than the number of *M. longifolia* individuals and to minimize potential errors introduced through plant identifications by different botanists (from the inventory and the present study), we aggregated the tree list to genera. We excluded all genera that were represented by only one individual in the inventory from the analyses and four individuals from two different species that could not be identified to genus level. The number of individuals per genus in the inventory (2 ha) was linearly extrapolated to the 78.3 ha area over which *M. longifolia* were found in the EBQB study area, calculated as a Minimum Convex Polygon around the GPS locations. The resulting number of individuals per genus was then used for further analyses (see *y*-axis labels in Fig. 2).

We used a Bayesian hierarchical regression model (*Gelman, Hill & Yajima, 2012*) for estimating the probability (Bernoulli distributed response, Logit link function; *McCullagh & Nelder, 1989*) of a tree hosting the liana, conditional on the genus of the tree (*i.e.,* genus was included as a grouping variable or 'random intercepts effect'; *Gelman & Hill, 2006*). We used the R statistical software environment (version 4.1.1; *R Core Team, 2022*) and especially the 'Stan' based package 'brms' (version 2.15.0; *Bürkner, 2017*; *Bürkner, 2018*; *Carpenter et al., 2017*) to conduct this estimation. As recommended by the *Stan Developers (2017)*, we used a truncated (at 0) Student-t distribution with 4 degrees of freedom as prior for the standard deviation of the hierarchically modelled differences (on the logit scale)

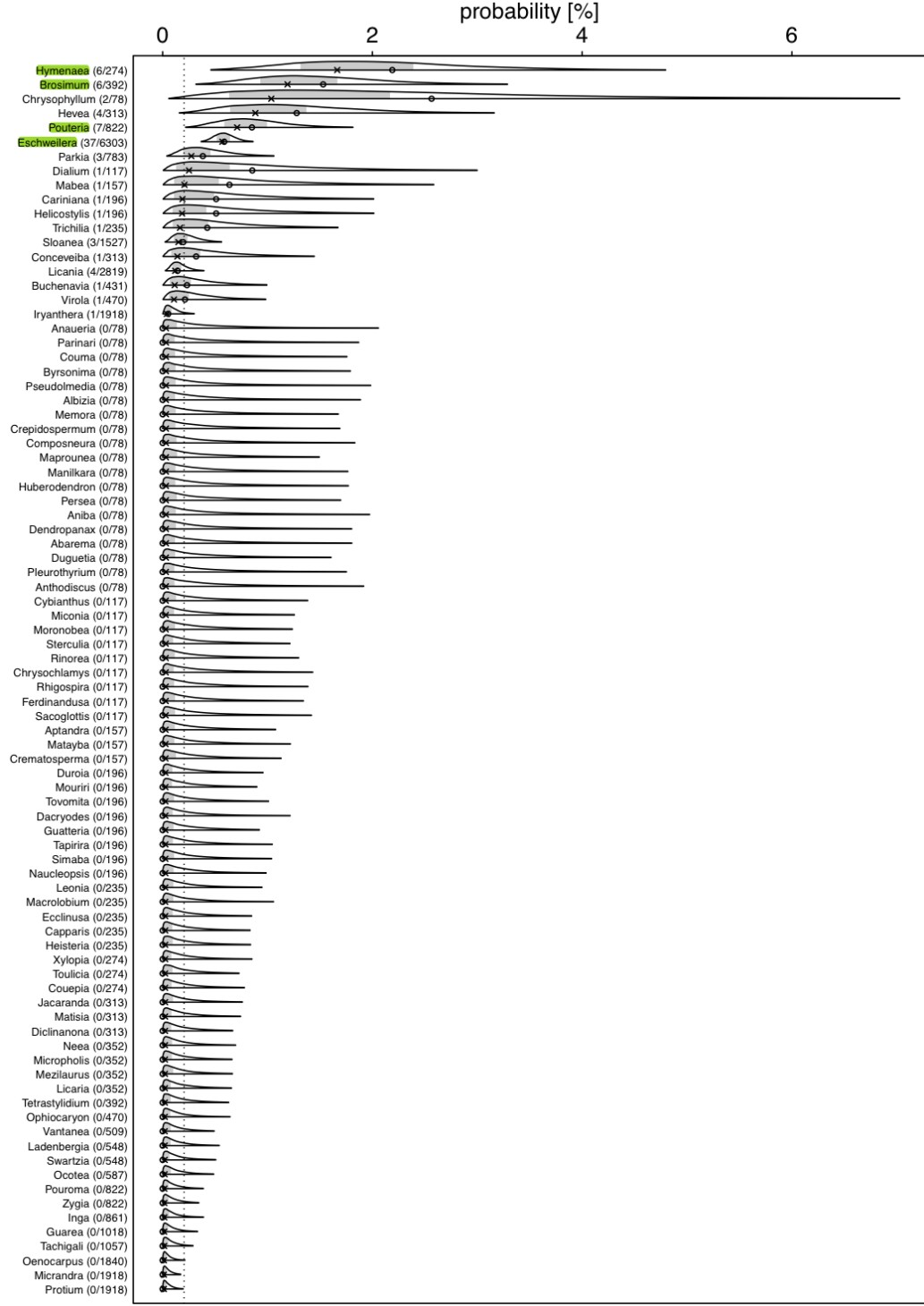

**Figure 2 Posterior density for probability (in %) of a tree hosting a *Marcgravia longifolia* across tree genera.** Numbers after genus names indicate number of infested trees per genus / extrapolated number of tree individuals per genus. Laying cross: posterior median; open dot: observed infestation probability; grey area: central 50% probability interval; plotting limits of densities: central 99% probability interval (all referring to the genera specific posterior distributions). Highlighted genera are those where the 99% credible intervals does not overlap with the overall probability of a tree being infested

between the genus-specific expected values. The Markov Chain Monte Carlo simulation of the posterior—as performed by brms—was run for 10 Markov chains, each with 5,000 iterations and a burn-in period of 2,500 iterations, and we set the Stan control parameters adapt_delta and max_treedepth to 0.95 and 20, respectively. The full R code reproducing our results is provided as Supplementary Material. As an overall test of whether genera differed in their probabilities to host a liana we inspected the 99% highest posterior density interval (determined using the function hdi of the package HDInterval, version 0.2.2; *Meredith & Kruschke, 2020*) of the estimated standard deviation (in link space) for the variability among genera. If this does not include the value of 0, the effect of genus on the probability of hosting a liana has a 0.99 posterior probability—given our data and model—to be larger than zero. The response was not overdispersed given the model (dispersion parameter assessed from an equivalent Generalized Linear Mixed Model: 0.101).

## RESULTS

During research on plant-animal interactions of *M. longifolia* and during studies on the ecology of small New World monkeys, we found a total of 100 adult *M. longifolia* individuals at our study site in northeaster Peruvian Amazonia. For 87 *M. longifolia* individuals, the host tree could be identified to species or genus level. Of the total of 88 tree genera with two or more individuals present in the tree inventory ($\geq$10 cm dbh) of *Dávila Cardozo & Ríos Paredes (2006)*, trees from 18 genera (20.5% of genera) were hosts; five other host genera were not represented in the inventory (Table S1). With 37 infested individuals, *Eschweilera* (Lecythidaceae) was the genus with the largest number of infested tree individuals; only one to seven individuals of other host genera were infested by *M. longifolia* (Table S1). Within *Eschweilera* 26 individuals (70%) belonged to *Eschweilera coriacea*, while the remainder belonged to three other species.

For several genera, the estimated probabilities of being infested were high and reached values of up to 1.66% (posterior median; Fig. 2). However, the 99% credible intervals for most of them were fairly wide. For *Eschweilera* (Lecythidaceae), *Pouteria* (Sapotaceae), *Brosimum* (Moraceae), and *Hymenaea* (Fabaceae), the credible intervals did not overlap with the overall probability of a tree being infested (Fig. 2; Table S2); also, for *Eschweilera* the credible interval was very narrow compared to other genera. For the majority of genera, infestation probability was estimated to be close to zero and associated with low uncertainly (*i.e.,* narrow credible intervals). The 99% credible interval for the variability (standard deviation of estimated genera specific deviations of their logit transformed infestation probabilities from the estimated logit transformed average infestation probability) among genera was estimated to range from 1.04 to 3.15 which is equivalent to an overall significant variation among genera with regard to their infestation probability (in case infestation probability would be similar in all species, particularly the lower but also the upper limit of this credible would be considerably closer to zero).

## DISCUSSION

Our results revealed a clear non-random component in the process that generates the infestation of hosts by *M. longifolia*. The estimated probability that a tree was infested varied considerably among genera. It was well above expected infestation probability in some and essentially zero in others. Although *Eschweilera* is the most common tree genus at our study site, it is infested more than would be expected from availability; trees from the second most common genus, *Licania*, are only very rarely infected. This strongly suggests that host choice may occur in *M. longifolia*.

The high probability of infestation of *Eschweilera* is surprising. It has been established that bark properties are influencing the probability of liana infestation: tree species with strongly exfoliating bark (flakiness) usually present lower levels of liana infestation since continuous debarking prevents lianas from growing successfully (*Carsten et al., 2002*; *Carrasco-Urra & Gianoli, 2009*; *Jiménez-Castillo & Lusk, 2009*). Similarly, trees with a smooth bark lack attachment points for climbers to grow on (*Carsten et al., 2002*; *Balfour & Bond, 1993*; *Campbell & Newbery, 1993*; *Putz, 1980*; *Putz, 1984*). Bark properties of *Eschweilera* trees vary between species (*Mori, Black & de Zeeuw, 1987*). The bark of *Eschweilera coriacea* (the most common *Eschweilera* species at EBQB with 68% of individuals of the genus) is strongly exfoliating (Fig. 1B), while the bark of other *Eschweilera* species is smooth, particularly in juvenile individuals (*Ribeiro et al., 1999*). These bark properties represent a challenge for liana infestation, but we do not know the strategy with which *M. longifolia* overcomes this challenge.

Since *M. longifolia* individuals are rooted close to the base of their host trees, seeds have to germinate next to the tree base to successfully infest a tree and grow up the trunk. Thus, the biased distribution of infestations over tree species could be a consequence of seed dispersal vectors of *M. longifolia* preferentially visiting *Eschweilera* trees for feeding, roosting, or sleeping. Fruits of *M. longifolia* are consumed by at least 41 species of birds, three species of primates, and >10 species of bats (*Tirado Herrera et al., 2003*; *Domingues Paciência, 2014*; *Willems, 2016*; *Gottstein, 2018*; *Thiel, 2021*). With the exception of coppery titi monkeys, *Plecturocebus cupreus*, none of these species is likely to exploit the hard-husked fruits of *Eschweilera*. Furthermore, *M. longifolia* and *Eschweilera* exhibit only a small overlap in their fruiting phenology: late ''dry'' to early rainy season in the former, and almost the entire rainy season in the latter (*Tirado Herrera et al., 2003*; *Flores et al., 2015*). Therefore, it is unlikely that seed dispersing frugivores—by targeting *Eschweilera* trees for feeding—disperse *M. longifolia* seeds selectively to the base of *Eschweilera* trunks. Alternatively, seed dispersal to *Eschweilera* trunks could occur if these trees were used for resting and sleeping. This can be excluded for the three primate species (*Leontocebus nigrifrons*, *Saguinus mystax*, *Plecturocebus cupreus*) whose sleeping and resting habits are well known (*Heymann, 1995*; *Smith et al., 2007*; *Muñoz Lazo et al., 2011*; Heymann et al. unpublished data); *Eschweilera* trees are not or very rarely used for sleeping and resting, respectively. In contrast to primates, sleeping and resting habits are largely unknown for the bird and bat species feeding on *M. longifolia* fruits, and thus they cannot be excluded as a factor creating directed seed dispersal to *Eschweilera* trees. However, we consider it more likely that the

large and diverse spectrum of seed-dispersing frugivores (birds, bats, primates) exploiting *M. longifolia* fruits creates a wide seed shadow, and that through unknown mechanisms *M. longifolia* is selectively recruited at and/or infesting *Eschweilera* trees.

In summary, our study provides tentative evidence for non-random host infestation in the Neotropical liana *M. longifolia*. It adds to a little-studied aspect of liana ecology and raises further questions on the mechanisms that may result in host choice by lianas and stimulate further studies, which include possible mechanisms of host selectivity.

## CONCLUSIONS

Our study provides preliminary evidence for non-random host infestation in the Neotropical liana *M. longifolia* and adds to the sparse literature on host selectivity in lianas. Future studies must identify the mechanisms of non-random host infestation, including traits of the liana and the hosts and properties of the microhabitat (*e.g.*, soils) around the hosts.

## ACKNOWLEDGEMENTS

We are grateful to Emérita R. Tirado Herrera from the Universidad Nacional de la Amazonía Peruana (UNAP) in Iquitos for her support. We are thankful to Felipe Mello and an anonymous reviewer for their constructive criticism which helped to improve the manuscript.

### Funding

This work was supported by grants from the Deutsche Forschungsgemeinschaft (DFG) to Eckhard W. Heymann (HE 1870/27-1) and to Katrin Heer (HE 7345-1). There was no additional external funding received for this study. The funders had no role in study design, data collection and analysis, decision to publish, or preparation of the manuscript.

### Grant Disclosures

The following grant information was disclosed by the authors:
The Deutsche Forschungsgemeinschaft (DFG): HE 1870/27-1, HE 7345-1.

### Competing Interests

The authors declare there are no competing interests.

Leibniz ScienceCampus Primate Cognition is the association of different institutions (University of Göttingen, Deutsches Primatenzentrum, Max Planck Institute for Experimental Medicine and others; see https://www.primate-cognition.eu/en/about-us/members.html). Therefore, it is an academic affiliation for Roger Mundry and Holger Sennhenn-Reulen (who worked on the data analyses while he was affiliated with the Leibniz ScienceCampus before getting an employment at the Nordwestdeutsche Forstliche Versuchsanstalt).

## Author Contributions

- Eckhard W. Heymann conceived and designed the experiments, performed the experiments, prepared figures and/or tables, authored or reviewed drafts of the article, and approved the final draft.
- Sarina Thiel conceived and designed the experiments, performed the experiments, authored or reviewed drafts of the article, and approved the final draft.
- Filipa Paciência conceived and designed the experiments, performed the experiments, authored or reviewed drafts of the article, and approved the final draft.
- Milagros N. Rimachi Taricuarima performed the experiments, authored or reviewed drafts of the article, and approved the final draft.
- Ricardo Zárate Gómez performed the experiments, authored or reviewed drafts of the article, and approved the final draft.
- Ney Shahuano Tello performed the experiments, authored or reviewed drafts of the article, and approved the final draft.
- Katrin Heer conceived and designed the experiments, authored or reviewed drafts of the article, and approved the final draft.
- Holger Sennhenn-Reulen analyzed the data, authored or reviewed drafts of the article, and approved the final draft.
- Roger Mundry analyzed the data, prepared figures and/or tables, authored or reviewed drafts of the article, and approved the final draft.

## Data Availability

The number and percentage of host tree genera and the number of host individuals per genus that are infested by *Marcgravia longifolia* are available in the Supplementary Table.

## Supplemental Information

Supplemental information for this article can be found online at http://dx.doi.org/10.7717/peerj.14535#supplemental-information.

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
