# Peer review of "Non-random host tree infestation by the Neotropical liana Marcgravia longifolia"

_PeerJ, doi:10.7717/peerj.14535_

## Round 0.1 · original submission · Minor Revisions

Critical changes

1. Both reviews highlight the need to be more specific about sampling & data collection procedure and the environment of the sampled area relative to the larger forest. These may affect the interpretation of how far the findings can be extrapolated.

2. Please take reviewer 2's statement that a null model for the regression is "crucial" as a suggestion that merits a thoughtful response rather than necessarily a requirement for acceptance.

3. Please be sure to include in your responses the points raised by Reviewer 1 that appear as comments on the PDF manuscript.

Optional changes & typos

4. Contra reviewer 1, if you wish to keep fig 1, that is fine, but I do agree with reviewer 2 that combining it with fig 3 might make sense.

5. In fig 2, please point out that the highlighted genera are the ones in the text at line 157 and/or the figure legend.

6. In the fig 3 legend, there is a typo in "host".

Reviewer 1 ·

Basic reporting

1. Title: The authors should try and avoid ambiguous words such as “non-random”. From the look of things, the authors should consider replacing this word with “selective” in the title or consider changing the title entirely. Similar changes should be made to this particular word throughout the text.
2. There are few grammatical errors that need to be corrected throughout the text. I have assisted with some of them.
3. I must commend the authors for bringing to fore the importance of conservation as regards lianas and contributing to liana research. However, I have suggested a few literature to be added to the text, especially in line 64.
4. Fig.1 is not that informative and might as well be removed. It seems the image only captured one of the authors climbing a tree.
5. The authors did well by providing raw data as well as the code used for their analyses

Experimental design

1. The research is very much within the scope of PeerJ, although the research questions are not well-defined and the justification for the study is superficial.
2. The methodology needs to be restructured and explained, most especially the sampling procedure used. It is not quite clear. In its present form, the methodology will be difficult to replicate and reproduce.

Validity of the findings

1. The findings are interesting with some interesting statistical supports
2. The conclusion is too brief. The authors should consider supplying more details to this section.

Annotated reviews are not available for download in order to protect the identity of reviewers who chose to remain anonymous.

·

Basic reporting

Dear Dr. Vision,
Thank you for the opportunity to review the manuscript “Non-random host tree infestation by the Neotropical liana Marcgravia longifolia”. The manuscript is an attempt to address an important and still unexplored ecological question of host preference by liana species. Studies of host preference by lianas is usually done using trees as focal species and quantifying the frequency of lianas or establishing phylogenetic networks. The present study, however, uses one single liana species to understand if the distribution of that species is non-random with respect to tree identity at the genus level.
The authors used a Bayesian hierarchical regression model to extrapolate the field data for the total area of the study site using data on tree genera abundance and abundance of M. longifolia host trees. The authors found that the liana species M. longifolia has a higher-than-expected probability of colonizing 4 genera of host trees out of the 88 tree genera present in the study site. The most abundant tree genera Eschweiler had particularly high occupation by M. longifolia, with about 40% of tree individuals carrying the liana species. Despite Eschweiler being the most abundant genus in the study site, the same patterns of high occupation by M. longifolia were not consistent with other abundant tree genus, which suggest a non-random occupation of some tree genera by M. longifolia.
The manuscript presents an interesting result for the unexplored area of host preference by lianas, however, the current version of the manuscript needs some critical clarifications and qualifications, which are detailed below.
1) the manuscript is not about liana specificity, as indicated by the introduction. Instead, the focus of the work is on whether one plant species, Marcgravia longifolia, which happens to be a liana, is non-randomly distributed among host trees (at the genus level). The lack of any replication at the growth-form level prevents any generalizations to growth form. Instead, the authors should limit their argument to the non-random distribution of Marcgravia longifolia (which is also interesting).
2) the manuscript also needs improvement in the readability and text flow. The current version of the manuscript has a considerable number of typos and unfinished sentences along the text. We highlight some along the text and give suggestions for improvement below. All the sections of the manuscript can be improved for clarity purposes. The introduction can be broken down in several paragraphs to better develop each key idea of the study’s background. I think that, despite the substantial improvement that the manuscript requires to better transmit the results, the manuscript has the merit of bringing a novel result using a different perspective from other studies on host preference by lianas.

Experimental design

3) The authors use the inventory of 2ha to extrapolate to 78ha. How generalizable the data from two hectares is compared to the whole forest? Perhaps there is a non-random association with Marcgravia and the few host species under the very specific conditions found in these 2 hectares. Or not. The point is that we don’t really know the scaling relationship between these 2 ha and the rest of the area.
4)The study also lacks a better description of the field site habitat distribution. Does the 2ha used in the survey have equal amounts of “terra-firme” forests and small swampy areas?
5) Another major concern is the lack of a comparable null model for the Bayesian hierarchical regression model. The analysis would be strengthened by a comparative null model showing that the asymmetric affinity to some tree genera over others is not due to random susceptibility of some genera over others. A valid null-model comparison is particularly important because there appears to be a positive relationship between tree infestation and the abundance of the host tree.

Validity of the findings

1)The manuscript is not about liana specificity, as indicated by the introduction. Instead, the focus of the work is on whether one plant species, Marcgravia longifolia, which happens to be a liana, is non-randomly distributed among host trees (at the genus level). The lack of any replication at the growth-form level prevents any generalizations to growth form. Instead, the authors should limit their argument to the non-random distribution of Marcgravia longifolia (which is also interesting).
2)The authors use an inventory of 2ha to extrapolate to 78ha, and the consequences of such extrapolation for generalization of the results has to be addressed.
3)A valid null-model comparison is critical to show non-random colonization of Marcgravia longifolia.

Additional comments

Below are more specific comments by section:
Abstract
Line 33 – “…or they exert host…” – This part does not match the rest of the phrase/paragraph. Exclude or rephrase for text consistency.
Lines 40-41 – “Of the total of 88 tree genera with two or more individuals…” – This paragraph sounds a bit confusing. Perhaps change for something like “From the total of 88 tree genera with..”.
Lines 43-44 – “These findings suggest that M. longifolia exerts host selectivity at least to some degree, but the mechanisms…” – Typo.
Lines 45-47 – I am not sure if this information adds much to the abstract. I think it would make more sense to end the abstract in the previous paragraph or to add a potential mechanism for the observed pattern, and not what potentially is not the mechanism (and save that for the discussion).
Introduction
The introduction needs to be improved for a better context of why the study is important. The current introduction has only two paragraphs, and the first paragraph contains too many half-developed ideas that can be split in new paragraphs to better develop each key idea necessary to understand the study background (Effect of lianas on trees -> consequences of host choice for forest dynamics -> empirical evidence for selection of hosts by lianas -> studies focused on the tree perspective …).
Lines 53-54 – Despite the focus of ecological studies on lianas being relatively recent and less developed than that of trees, there has been a considerable increase in liana ecology studies in the past 20 years. Also, this statement breaks the connection between the previous and following paragraphs. I would suggest removing the statement or rephrasing it to take in account for all the recent efforts to include lianas in ecological studies.

Line 56 – You can make a new paragraph starting with “Infestation patterns have been mainly addressed…”

Lines 58-59 – “Fast-growing tree species that are tall as adults are less likely to support lianas (van der Heijden et al. 2008)…” – Suggestion of citation Visser et al (2018) Journal of Ecology. The authors discusses the effects of lianas on fast-growing vs slow-growing species.

Line 59-60 – in “and trees with larger diameter at breast height (dbh) support more lianas and a higher liana biomass” – English change suggestion “and trees with larger diameter at breast height (dbh) support higher liana density and biomass”.

Lines 63-64 – The question whether lianas prefer or avoid host trees has been addressed a couple of times. Some studies include Sfair et al (2016) Tropical Ecology; Sfair et al (2017) Austral Ecology; Zulqarnain et al (2016) Perspectives in plant ecology, evolution and systematics.

Line 82 – in “Therefore, set out to systematically test the hypothesis..”. You meant “…, we set out to..”?

Methods

Line 93 – “…with a peak from February to May and a trough between August…” – Something is missing here.

Data collection
This part of the method needs to be improved to give the reader a better understanding of the raw data used for the statistical model. How variable were the tree host individuals within each species? Was there variation on diametric or height distribution of tree individuals? Were some species taller than others within genera groups? I would suggest perhaps including on table 1(or maybe another table) some ecological and/or demographic distribution of the species within each genus, to have an idea of the amount of variation for each genus group. How different are traits within species of the same genus of host species? Do all species within the genus Eschweilera have the same bark traits, for instance? You talk about those points in your discussion, but that could be introduced earlier in the methods.
What is the diversity of lianas in the study site? Is there any other specific reason for choosing M. longifolia beyond their easily recognizable flowers and fruits?
Line 101-104 – When is the flowering/fruiting period of M. longifolia? Again, you talk about that in the discussion, but it can be introduced earlier, at least for M. longifolia, in the methods.

Data analysis

118-120 – Why did you use convex polygons to calculate area? I would suggest a short paragraph discussing other studies that used such methods and the potential implications of an extrapolation of 2ha in 78ha.

122-141 – Is M. longifolia specially targeting the tree genera or are these tree genera only more susceptible to lianas in general? This is a major concern about the study analysis. It is crucial that you incorporate a comparative null model to compare with your model’s posteriors and show that the asymmetry affinity observed for some genera is actually from M. longifolia targeting more of them than the other host genera. Since most of the inference that M. longifolia has a non-random host preference is being made from the study’s analysis, a comparative null model is critical.

You need a better connection between the data collected in the field and your modeling strategy. As I commented before, you have to better describe the data used for your model, using perhaps a table to better organize and describe the data used as your priors, and connect that in a smoother way with the construction of your statistical model.

Results

144-147 – The first two sentences of the first paragraph would fit better in the methods section and can be included in the description of the data collected, as suggested above. The results can be more direct to the point, starting in “Of the total of 88 tree genera with 2 or more individuals…”. (Also suggest changing the symbol > for the suggested version above).


155-156 – In the sentence “For several genera, the estimated probabilities of being infested were quite high and reached values of up to 1.66% (posterior median; Fig. 2)” – Explain better what you mean by “quite high infestation” and what the 1.66% represent (% of infested individuals?).

Discussion

186-188 – for the statement “These bark properties represent a challenge for liana infestation, but we do not know the strategy with which M. longifolia overcomes this challenge.” - Could the adventitious roots used by M. longifolia be a mechanism to facilitate climbing and overcome the bark trait challenge? You may include in our discussion how the use of adventitious roots may be a factor (or not) helping in the non-random host choice observed.

209-212 – Despite providing evidence that preference for Eschweilera by dispersing frugivore is unlikely to be the mechanism for differential occupation of that species by M. longifolia, it would be interesting if the authors include one or two potential alternative mechanisms that would be driving the increased pattern of occupation of Eschweilera by M. longifolia.

Regarding the dismissal of dispersion as a mechanism for the pattern observed, how can we make sure that the pattern observed is not due to the preference of non-primate dispersal animals to the tree species instead of preference of the lianas to the host trees? I think the authors should develop a better paragraph about the potential mechanisms that are shaping the observed pattern of M. longifolia occupation (instead of only highlighting the main unlikely mechanism).

Figure 1 and figure 3 may be combined in only one figure.

I hope our comments are useful and can help to improve your manuscript.

---

## Round 0.2 · accepted · Accept

I appreciate the thoughtful responses to the reviewer's comments. I have assessed the revision myself and am happy with the current version, provided the authors address the following two details before the MS is finalized:

1. In the response to editor's comment 5, the authors state "We added the following text to the legend: Highlighted genera are those where the 99% credible intervals does not overlap with the overall probability of a tree being infested." but this text is not actually present in the revision.

2. On line 177 in new version it states: "For several genera, the estimated probabilities of being infested were high". The authors deleted "quite" but not "high", again contra to the response letter. The authors may fix the wording if that was not the intent (but I have no objection to the present wording in the MS).